# The Role of Zinc and Copper in Platelet Activation and Pathophysiological Thrombus Formation in Patients with Pulmonary Embolism in the Course of SARS-CoV-2 Infection

**DOI:** 10.3390/biology11050752

**Published:** 2022-05-14

**Authors:** Monika Szewc, Agnieszka Markiewicz-Gospodarek, Aleksandra Górska, Zuzanna Chilimoniuk, Mansur Rahnama, Elżbieta Radzikowska-Buchner, Karolina Strzelec-Pawelczak, Jarosław Bakiera, Ryszard Maciejewski

**Affiliations:** 1Department of Human Anatomy, Medical University of Lublin, 20-090 Lublin, Poland; monikaszewc@umlub.pl (M.S.); agnieszkamarkiewiczgospodarek@umlub.pl (A.M.-G.); ryszardmaciejewski@umlub.pl (R.M.); 2Student Scientific Group, Department of Family Medicine, Medical University of Lublin, 20-032 Lublin, Poland; zuzia.chil@gmail.com; 3Department of Oral Surgery, Medical University of Lublin, 20-093 Lublin, Poland; mansur.rahnama@umlub.pl; 4Department of Plastic, Reconstructive and Maxillary Surgery, CSK MSWiA, 02-507 Warszawa, Poland; elzbieta.radzikowska@gmail.com; 5Resident of Neonatology Department SPSK nr 1 in Lublin, 20-081 Lublin, Poland; k.strzelec89@wp.pl; 6Department of Laboratory Diagnostics, Regional Specialist Hospital in Lublin, 20-718 Lublin, Poland; jbakiera@wp.pl; 7Institute of Health Sciences, The John Paul II Catholic University of Lublin, 20-708 Lublin, Poland

**Keywords:** SARS-CoV-2, COVID-19, pandemic, thrombus, embolism, zinc, copper, vascular changes

## Abstract

**Simple Summary:**

Recent studies associated COVID-19 with coagulation dysfunction and increased risk of thromboembolism. Consequently, pulmonary embolism (PE) development and higher odds of mortality were observed in this group of patients. CRP levels, which are considered a sensitive marker of systemic inflammation, were linked to the higher rates of PE and immunologic processes associated with atherosclerosis development and thrombus formation. Several microelements, including zinc and copper, were shown to present anti-inflammatory and anti-oxidative characteristics. Due to these properties, and they’re in-fluence on platelets activation and thrombus formation, zinc and copper should be evaluated as a potential therapeutic option during COVID-19.

**Abstract:**

To date, many studies have proved that COVID-19 increases the incidence of thrombus formation and coagulopathies but the exact mechanism behind such a disease outcome is not well known. In this review we collect the information and discuss the pathophysiology of thrombus formation in patients with pulmonary embolism in the course of COVID-19 disease and the role of zinc and copper in the process. Supplementation of zinc and copper may be beneficial for COVID-19 patients due to its anti-inflammatory and anti-oxidative properties. On the other hand, excess of those microelements in the organism may be harmful, that is why marking the level of those micronutrients should be done at first. We also propose further investigation of diagnostic and therapeutic options of zinc and copper in course of COVID-19 thrombus formation to their potential in patient care, with particular emphasis on the dosage and the duration of their misbalance.

## 1. Introduction

Zinc and copper are microelements that play a significant role in fundamental physiological processes. They were also proven to be crucial for the human immune system functioning [1,2]. Both present antiviral activities. While zinc reduces the viral replication and improves immune cell function, copper neutralizes infectious DNA and RNA viruses and activates autophagy [3,4,5]. Zinc supplementation may also shorten the duration of respiratory tract infections [6]. It was also suggested that zinc may affect platelets activation and blood clotting which is an important aspect in case of COVID-19 symptoms [7,8].

COVID-19, a viral respiratory illness, has affected millions of people worldwide and significantly increased the risk of morbidity and mortality [9,10]. The respiratory symptoms are the main feature of the disease, but evidence showed that COVID-19 is also associated with coagulation dysfunction which predisposes to increased risk of thromboembolism. There are multiple reports that demonstrate increased rates of thromboembolic events in patients with COVID-19 [11,12,13,14,15,16,17,18,19,20]. The rate of thromboembolism as a complication of COVID-19 reported in the literature is varied. Some studies have showed that thromboembolism rates in the range of 20–30% [12,21,22] while others have claimed rates as high as 40–70% [15,23]. The presence of hypercoagulation complications been noted to correlate with a more severe course of the disease and is associated with poor prognosis involving potentially death [24]. There are studies that prove that thromboembolism rates of COVID-19 were high and COVID-19 patients who developed thromboembolism were at a significantly higher (74%) odds of mortality compared to those who did not [25]. Thrombosis in segmental and subsegmental pulmonary arterial vessels may lead to death even despite the application of anticoagulants [26]. There are a lot of factors that can be associated with higher rates of pulmonary embolism during hospitalization, for example D-dimer and CRP levels [27]. In our opinion it is curtailed to understand the exact pathophysiological mechanism of formation of the thrombus in patients with pulmonary embolism because it is one of the most common complications in the course of SARS-CoV-2 disease. Additional aspect of the influence of zinc and copper on platelet activation and thrombus formation may be the base for further therapeutic solutions in COVID-19 treatment.

## 2. Relationship between Zn^2+^/Cu^2+^ and CRP Levels and Platelet Activation

Recent studies investigated the correlation between metals and CRP, whose levels were positively associated with copper plasma concentrations. This indicated that copper’s higher level may enhance the genetic predisposition to increased CRP level. In this case, a new perspective on the role of copper exposure in triggering inflammation was presented. In addition, higher zinc plasma concentration weakened the interaction between copper and CRP [28]. Mousavi’s systematic review and meta-analysis confirmed an inverse association between zinc supplementation and plasma CRP concentrations [29]. Previous studies also highlighted the significant correlation between CRP and zinc and copper levels [30,31].

Above mentioned CRP plays a role of an acute-phase inflammation biomarker and a predictor of cardiovascular events. Once it dissociates to monomers, mCRP acquires new proinflammatory functions including the ability to promote thrombosis [32,33,34]. Most importantly, mCPR can induce platelet adhesion, activation, and thrombus formation. Due to the CRP presence on the platelet surface and within the thrombus, the interaction between platelets and CRP was suggested [35,36]. In addition, the use of immunohistochemical analysis led to the discovery of CRP in atherosclerotic lesions with the presence of VSMC and macrophages [37,38]. As CRP expression in active plaques is increased, mechanical or spontaneous plaque rupture of the lesions could lead to platelet aggregation and thrombus formation enhancement [35,39].

mCRP was proven to upregulate platelet P-selectin, which is responsible for platelet–platelet and platelet–leukocyte aggregates stabilization. Furthermore, this could result in thrombus growth promotion [35,40]. As mentioned before, mCRP is present in growing platelet aggregation where it stimulates further platelet deposition. However, the study showed that the platelet receptor glycoprotein IIb-IIIa (GPIIb-IIIa) blockage prevents the pCRP dissociation to mCRP. Thus, a reduction in platelet deposition at the arterial wall was obtained [36].

It was also indicated that mCRP enhanced platelet procoagulant activity which was expressed as an increased fraction of annexin V-positive platelets [41]. Annexin V alone is an important marker for platelet activation as it binds to the plasma membrane and actin-based cytoskeleton in activated platelets [42]. Interestingly, mCRP was shown to enhance annexin binding to blood platelets and platelets microparticle formation. Furthermore, GPIb-IX-V complex, which is a potent platelet receptor, was shown to play the role of the mediator in the interactions of mCRP with blood platelets. The fact that one of its major ligand-binding subunits which is glycoprotein Ibα, was directly bound by mCRP supports the role of the complex as a mediator [41].

All things considered, increased CRP concentrations following higher copper plasma levels could enhance CRP role as an agonist in platelet activation. In addition, zinc has been shown to play an important part in hemostasis [43]. Exogenous Zn^2+^ may influence platelet activation and initiate its aggregatory mechanisms as it accesses the platelet cytosol and modifies the activity of different enzymes. It was proven to interact with protein kinase C (PKC) which results in granule release and integrin αIIbβ3 activation. Additionally, Zn^2+^ inhibits cytosolic protein phosphatases, activates protein tyrosine kinases and is responsible for adenylate cyclase inhibition [7]. The precise process of platelets activation caused by Zn^2+^ is presented in Figure 1.

## 3. Vascular Changes versus Metals and Metalloproteins

SARS-CoV-2 virus infection in humans is closely associated with a broad spectrum of clinical manifestations including mild upper respiratory symptoms but may also result in severe viral pneumonia. It was suggested that SARS-CoV-2 infection may impair microcirculatory function by causing endothelial cell injury and inducing endotheliitis [47]. Importantly, vascular complications caused by venous thromboembolism or atherosclerosis were the leading causes of death in this group of patients [48,49]. However, zinc with its anti-inflammatory and anti-oxidative properties could plausibly improve clinical outcomes in patients with COVID-19 infection [49,50,51]. In addition, zinc plays an important role in wound healing as it regulates cell membrane repair, tissue re-epithelialization, angiogenesis, and other phases of this process [52]. However, zinc is also considered a platelet agonist and was proven to induce ROS production in platelets. Thus, zinc pharmacological manipulation could allow to moderate oxidative stress and influence thrombus formation [53].

Patient difficulties with diagnosed COVID-19 infection are largely related to respiration, requiring mechanical ventilation support. Peripheral lung ground-glass opacities have been observed on CT scan which meet criteria for acute respiratory distress syndrome ARDS [54,55]. In addition to the characteristic symptoms of COVID-19 vascular lesions are closely associated with the disease. Moreover, many patients have elevated d-dimer levels and interestingly skin lesions that suggest thrombotic microangiopathy [56]. Disseminated intravascular coagulation and large vessel thrombosis have been associated with multiple organ failure, including during COVID-19. In addition to elevated d-dimers we can demonstrate a reflection of cardiovascular involvement by increasing the release of highly sensitive troponin and natriuretic peptides which are diagnostically important (particularly when we see an increase level) along with associated cytokines such as interleukin-6 (Il-6) [56].

In addition, a study based on optical coherence tomography angiography (OCTA) has recently been published that showed disruption of micro vessels within the retina [57]. Another study conducted by Jidigam et al. aimed at postmortem analysis of the fundus demonstrated vascular anomalies consistent with ocular vein obstruction and reduced capillary density [58].

The above reports suggest a close association between mortality, elevated d-dimer values and prothrombotic syndrome. Vascular complications also occur in other organs as mentioned above. A detailed analysis of currently available data, medical, laboratory and imaging reports on COVID-19 confirms that not all symptoms and diagnostic test results can be associated with impaired lung ventilation. The pathological process begins in the lungs with impaired perfusion, but then spreads to other organs with severe disease. The virus causes microvascular damage causing microthrombotic lesions in lung capillaries and organs leading to microthrombosis and embolism [59]. The above evidence clearly indicates that early thromboprophylaxis, systematic monitoring of laboratory parameters, performing imaging studies (CT) and early pharmacological (anticoagulant) treatment can improve the patient’s condition and prevent potential vascular complications.

## 4. Oxidative Stress and Inflammation

Oxidative stress can be defined as an imbalance between the intensity of oxidative processes that induce the formation of reactive oxygen species (ROS) and the antioxidant defense system [60]. ROS can cause oxidation of fats, protein, DNA and consequently contribute to tissue damage. Toxic products formed by oxidation have a cytostatic effect on the cell, causing damage to cell membranes and leading the cell to death by apoptosis and necrosis [61]. During normal, physiological cellulat fynctions, ROS are needed to maintain normal, efficient bilogical functioning. However, the role of ROS is closley coordinated through redox regulation, redox sensing, and redox signaling. Under disturbed physiological conditions, oxidative stress occurs due to uncontrolled ROS production, in which redox signaling is impaired and contributes to damage at the molecular level [62]. On the other hand, the state of normal cellular equilibrium is ensured by antioxidant enzymes such as superoxide dismutase, catalase, S-glutathione transferase, and other substances such as glutathione, vitamin E, C and A [60]. The above-mentioned compounds are responsible for removing excess ROS from cells and thus translate into the maintenance of normal homeostasis.

Oxidative stress begins with excessive production of ROS by the body and an imbalance of antioxidants [63]. This condition is associated with an imbalance of redox homeostasis in the body. As mentioned earlier, ROS can cause damage at the molecular level by reacting with nucleic acids, proteins, carbohydrates, and lipids. Every organism in contact with oxygen during evolution has developed defense metabolic mechanisms to protect their integrity against free radicals, such as [60,64]:-Reactions involving compounds that quench excited molecules (carotenoids, vitamin E)-Non-enzymatic mechanisms—uric acid, bilirubin, glutathione, pyruvate, ubiquinone (coenzyme Q), transferrin, polyamides, transition metal ions, metalloproteins-Enzymatic mechanisms: superoxide dismutase (SOD)—catalyzes the superoxide anion radical dismutation reaction, catalase (CT)—catalyzes the hydrogen peroxide dismutation reaction, glutathione peroxidase (GPx), ceruloplasmin, glutathione S-transferase (GST), secretory phospholipase group A_2_ (sPLA_2_)-Heat shock proteins (Hsps)—a large family of molecular chaperones that can be divided into two groups: first—small, ATP-independent Hsps with molecular weights from 8 to 28 dKA e.g., ubiquitin, and second—large, ATP-dependent Hsps with molecular weights from 40 to 105 kDA; a group of proteins whose expression increases when cells are exposed to stress factors (e.g., osmotic stress, heavy metals).-It is well known that viral infections can alter the redox system increasing oxidant species and reducing antioxidant molecules. Varga et al. reported that there is a link between ROS, endothelial damage, and inflammation, and this above mechanism also occurs during the course of COVID-19 [47].

### 4.1. The Role of Zn^2+^/Cu^2+^ in the Development of Oxidative Stress

Copper and zinc are trace elements that play an important role in many vital functions and metabolic processes of the human body. In venous vessels Zn^2+^/Cu^2+^ are involved in the formation of oxygen free radicals, catalysts of the Haber-Weis reaction described in 1876 [65]. Copper-containing proteins are involved in oxygen transport, electron transport, and catalyze oxidation and reduction reactions. In addition, if the levels are too high, they can contribute to an increased incidence of DNA mutations [66]. On the other hand, zinc has a role in regulating glutathione peroxidase and metallothionein expression, a cofactor role for superoxide dismutase. Additionally, zinc competes with copper and iron in the cell membrane by inhibiting the enzyme NADPH oxidase, which is associated with a reduction in chronic inflammation [67,68]. This enzyme promotes the conversion of two peroxide radicals to hydrogen peroxide and molecular oxygen, reducing the toxicity of ROS because it converts a highly reactive species to a less harmful one [67]. Therefore, it is important to maintain optimal zinc concentrations which are essential for the proper functioning of antioxidant defenses. 

Another known antioxidant activity of zinc is its effect on the expression of glutamate-cysteine ligase. This has a two-pronged effect: direct—(via glutathione) and indirect (as a cofactor of glutathione peroxidase) [69]. Additionally, it is a potent inducer of metallothionein where under physiological conditions it connects while under oxidative stress it is released from its complex with metallothionein and is redistributed in cells to ultimately exert antioxidant effects [70]. The last proposed mechanisms it that zinc binds to sulfhydryl groups in proteins while simultaneously protecting them [69]. This is another option for how zinc plays an antioxidant role while helping to inhibit the production of ROS. 

### 4.2. Vascular Complications and Their Relationship with CRP Levels 

C-reactive protein (CRP) is an acute phase protein that belongs to the family of calcium-dependent ligand-binding plasma proteins [71]. It is considered an exquisitely sensitive marker of systemic inflammation, tissue injury and infection as its blood concentration increases rapidly in these conditions [72]. The circulating pentameric form of CRP (pCRP) consists of five identical non-glycosylated globular subunits of 206 amino acid residues. Because of its structure, it is classified as a member of the pentraxin family [72,73].

The synthesis process that takes place in the liver is regulated by proinflammatory cytokines including main inducer interleukin (IL)-6 and may be enhanced by IL-1β, and tumour necrosis factor (TNF) [74,75]. What is worth mentioning, vascular smooth muscle cells (VSCM) infection or trauma may contribute to (IL)-6 release in response to atherosclerosis [76]. Extrahepatic CRP synthesis was also reported in other tissues such as adipose tissue in response to inflammatory cytokines which suggests a correlation between obesity and vascular inflammation [77]. Moreover, CRP presents different functions compared to pCRP when it dissociates to monomers. Likely, previously mentioned dissociation into mCRP is crucial for the inflammatory processes associated with atherogenesis [32]. mCRP was also shown to have prothrombotic properties such as platelet activation or thrombus formation [35,78].

CRP is not only considered an important predictor of future cardiovascular events but also, it was indicated that it may participate in atherogenesis and cardiovascular disease progression. In addition, CRP elevated concentrations were associated with increased risk of venous thromboembolism (VTE) that comprises PE [79]. Several studies also showed a link between atherosclerosis and VTE suggesting that the first mentioned may play a role in thrombus formation in PE patients [80,81,82].

Importantly, it has been associated with the immunologic process that gives rise to vascular remodelling and plaque deposition [33,34]. Regarding atherogenesis, growing and unstable atherosclerotic plaque is crucial for inducing ischemic events as it leads to thrombus formation [83]. Thus, CRP may contribute to this process by activating the complement system, inducing apoptosis, activating vascular cells, and taking part in leukocyte recruitment, lipid accumulation and platelet aggregation [84].

It has been shown that CRP activates inflammatory cells which results in monocyte adhesion and transmigration into the vessel wall. In addition, by binding the phosphocholine of oxidized low-density lipoprotein (LDL), CRP promotes macrophages uptake of LDL to form foam cells and develop inflammation [85,86]. Furthermore, pCRP induces monocyte polarization to M1 and phenotype conversion, which leads to monocyte recruitment to adipose tissue and atherosclerotic lesions [87].

CRP may also affect plaque stability by activating nuclear factor κB (NF-κB). This could increase endothelial cell adhesion molecules expression including vascular cellular adhesion molecule-1 (VCAM-1), vascular E-selectin and monocyte chemoattractant peptide (MCP-1) [88,89]. The loss of pentameric symmetry and rearrangement into mCRP may result in peroxynitrite-mediated activation of NF-κB and activator protein-1 (AP-1) which leads to increased expression of IL-8 and CD11b/CD18 adhesion molecules on human neutrophils. This correlation shows a link between mCRP bioactivity and enhanced inflammatory response due to increased neutrophil adhesion to the activated endothelial cells [90,91].

Moreover, it has been shown that CRP is directly involved in the mechanisms of atherosclerotic plaque remodelling and destabilization because of its ability to upregulate metalloproteinases (MMP) 1, 2, and 9 expressions [92,93]. In addition, CRP was also suggested to take part in the neovascularization of intima in vulnerable plaques which could contribute to its destabilization and progression of atherosclerosis [94]. Interestingly, CRP is proven to enhance vascular endothelial growth factor-A (VEGF-A) expression and to increase the activity of MMP-2 which may contribute to angiogenesis promotion in inflammatory conditions [95].

In addition, CRP induces endothelial dysfunction by inhibiting endothelial nitric oxide synthase, increasing production of the endothelin-1, which is a potent endothelium-derived constrictor and by altering renin-angiotensin system activation. Above mentioned CRP actions lead to vasoreactivity impairment [96,97].

## 5. The Role of Inflammation in Relation to COVID-19 and Potential Complications

The SARS-CoV-2 virus is transmitted by airborne droplet, contact and faecal-oral route [98]. The course of SARS-CoV-2 infection was first described by Wang et al. who also reported the characteristics of the infection manifested as abnormalities in laboratory results and clinical symptoms [99]. Virions suspended in the form of an aerosol usually survive up to 3 h in an environment with an average temperature of 21–23 °C, on the surface of texture e.g., paper around 24 h, and up to 72 h on the surface of object which are made of plastic or stainless steel [100]. Primary viral replication is believed to take place in the musical epithelium of the upper respiratory tract and downstream in the lower respiratory tract and gastrointestinal mucosa, leading to mild viraemia [101,102]. If infection occurs, the incubation period is 2 to 14 days (average incubation period 4–5 days) before the first clinical symptoms appear. Symptoms appeared in approximately 97% of patients within day 11 [103,104]. Infection and disease can be divided into 3 consecutive phases: first—asymptomatic phase, with or without the possibility of virus identification; second—mild symptomatic phase with upper respiratory tract involvement and third—severe, potentially fatal disease with hypoxia, “ground glass” infiltrates in the lungs, and progression to acute respiratory distress syndrome (ARDS) with high level viral load [105]. Current evidence suggests that a subset of patients with diagnosed severe COVID-19 may have cytokine release syndrome. These patients have been reported to have high titers/concentrations of pro-inflammatory cytokines and chemokines which are closely associated with inflammation and extensive lung injury [106]. In a recent study, COVID-19 was shown to promote coagulopathy through pro-inflammatory cytokines or activation of immune cells in response to alarm signals such as: tissue damage, complement activation, and/or formation of autoantibodies and immune complexes that may enhance coagulation. The above mechanisms may result in thrombosis, pulmonary embolism and may ultimately lead to death. Figure 2 illustrates the course of SARS-CoV-2 infection and its potential consequences depending on the severity of the course.

SARS-CoV-2 virus infection causes gradual destruction of lung cells. Damage to the structure of these cells induces a local immune response and thus results in the recuitment of macrophages, monocyte-derived cells arising from the bone marrow. The above cells i.e., macrophages and monocytes in response to infection release cytokines and stimulate the immune response to produce T and B lympgocytes [109]. In most cases the process of activating the local immune response is sufficient and leads to the disappearance of the disease process. However, when any immune disorders develop, secondary complications including serious pneumonia may occur. Higher plasma levels including Il-2, Il-7, Il-10, granulocyte colony-stimulating factor (G-CSF), interferon gamma inducible protein-10 (IP-10), monocyte chemoattractant protein-1 (MCP1), macrophage inflammatory protein 1α (MIP1α) and tumor necrosis factor (TNF) were found in patients who had severe COVID-19 disease and thus required intensive hospital care [9,110]. In addition to local damage the cytokine storm also has far-reaching effects thought the body. Elevated TNF levels may contribute to septic shock and thus to multiple organ failure. This can result in myocardial damage and circulatory failure in some patients—especially those over the age of 60 with comorbidities [109,111]. The transition between the innate and adaptive immune response is critical to the clinical progression of SARS-CoV-2 infection. At the pivotal point there are two situational options—there’ll be the development of a protective immune response or there’ll be the development of an enhances inflammatory/severe inflammatory response [112]. An immune response that depends on T lymphocytes, which are involved in helping B lumphocytes, which in turn are targeted to produce specific neutralizing antibodies and CD8 cytotoxic cells tkat have the ability to eliminate infected cells. In comparison, the pathological immune response is incapable of inhibiting viral replication and eliminating infected cells. This leads to an enhanced inflammatory response most likely leading to a cytokine storm manifesting as ARDS and systemic consequences such as disseminated intravascular coagulation.

In addition to the frequently described systemic complications and ARDS, significant procoagulant events including life-threatening pulmonary embolism (PE) are worth noting [113]. Pulmonary embolism is a disease in which the lumen of a pulmonary artery or its branches is narrowed or closed by embolic material. The function of these vessels is to supply blood from the heart to the lungs. Pulmonary embolism can be caused by a variety of embolic material. In most cases it is a thrombus or a lump of clotted blood. Its formation usually occurs because of blood clotting, the so-called Virchow’s triad i.e., predisposition to thrombosis, injury to the vessel wall and slowing of blood flow [80]. Figure 3 shows selected risk factors that predispose to pulmonary embolism. 

Currently, little is known about the epidemiology and occuring pathophysiological mechanisms underlying PE closley associated with COVID-19 due to the lack of large prospective studies in this context. COVID-19 complicated by acute PE may mark a turning point in the patient prognosis by the additional hypoxemia or hemodynamic collapse, which lead to intensive care unit (ICU) admission and mechanical ventilation [117]. Current recommendations recommend prophylactic anticoagulant therapy for all patients admitted to the hospital with severe course of COVID-19 [118]. Unfortunately, despite the applied anticoagulant prophylaxis the risk of PE is still high. 

## 6. Therapeutic Modifications of Zn^2+^/Cu^2^ Levels in Relation to Pulmonary Embolism and Patients with COVID-19 

Zinc (Zn^2+^) is an microelement mandatory for support a variety of fundamental physiological processes. Proper functioning of the immune system also depends on the level of zinc in the body [1]. Zinc deficiency is a common condition in elderly people and those with chronic diseases, due to low zinc intake or malabsorption. Those two groups are also in increased risk for severe COVID-19 outcomes [119]. Zinc demonstrates antiviral activity by equalizing immune responses and reducing the ability of viruses to multiply or by improving immune cell function that counteract viral infections [1,3]. Moreover, zinc supplementation remarkably shorten the course of respiratory tract infections caused by viruses [6]. Studies in vitro results indicate that low zinc levels favor viral expansion in SARS-CoV-2 infected cells [120,121]. Zinc supplementations enhance cytotoxicity when used in vitro [122]. On the other hand, long-term zinc supplementation can cause copper deficiency with subsequent reversible hematologic defects and potentially irreversible neurologic manifestations [123,124]. Moreover, zinc deficiency can be the cause of more tissue damage, such as damage seen in lungs during pulmonary embolism where it decreases the action of pulmonary surfactant [125,126,127].

It was proved that zinc is involved in blood clotting which is important in conjunction with pulmonary embolism during SARS-CoV-2. When the level of zinc in blood is faulty, the blood clots can be formed. Additionally, the work of heparins also depends on zinc level. Moreover, clots formed in the presence of zinc are less stiff than those formed in its absence, are resistant to perturbation by mechanical forces and are more porous [8]. Another potential zinc related therapeutic plan affects the expression of ACE2 receptors, which are essential to SARS-CoV-2 infection for the entrance of target cells [128]. ACE2 is expressed on type 2 pneumocytes and requires zinc to work properly. In conclusion, zinc generate thrombolytic processes during COVID-19, fibrin degradation, blood flow, ROS generation and reperfusion after thrombolysis [129].

Zinc supplementation may be additional procedure during the treatment for COVID-19 patients to lower the recovery time or even reduce mortality [130]. However, there are also studies that indicate that zinc supplementation did not have any beneficial impact on the course of COVID-19 evaluated as survival to hospital discharge and in-hospital mortality [131]. That is why further investigation is required.

Copper (Cu^2+^) is a crutial microelement for the function of the human immune system [2]. Cu is involved in the functions of immune system [132]. Cu has the potent capacity to neutralize infectious single- or double-stranded DNA and RNA viruses [4].

Cu deficiency is rare and usually only happens in seriously ill people receiving parenteral nutrition that lacks Cu [133]. Cu deficiency symptoms in human include deficiencies in white blood cells, bone and connective tissue abnormalities, and immune reactions [132]. There is study on mice fed the Cu-deficient, compared with Cu-supplemented diet. Cu-deficient diet was found to have significantly increased activated partial thromboplastin time and prothrombin time [134]. While high copper levels can be poisonous the Cu levels must be maintained optimally.

Based on available data, we hypothesize that the high level of copper in plasma improve functioning of immune system and could be used in preventic and therapeutic procedures in COVID-19 patients [135]. Another beneficial aspect of copper is the activation of the antioxidant defensive system [136] which leads to the protection against oxidative cellular injury [137]. Additionally, the antipathogenic effect of copper is the activation of autophagy to maintain the cell’s antiviral effect [5]. The understanding of copper dosage opens up new perspectives to therapeutic copper administration in people infected with SARS-CoV-2. However, toxicity of copper should not be forgotten so the estimation of copper dose and the duration of copper misbalance are also very important aspects.

## 7. Conclusions

According to the currently available literature, it was suggested that COVID-19 is closely connected with abnormal coagulation and can lead to thromboembolism and PE in consequence. Factors such as CRP levels can be linked to higher rates of pulmonary embolism during hospitalization. Interestingly, CRP has been also associated with the immunologic process that gives rise to vascular remodeling and plaque deposition which are crucial for atherosclerosis development and thrombus formation. Due to zinc and copper anti-inflammatory and anti-oxidative properties, their supplementation could plausibly improve clinical outcomes in patients with COVID-19 infection. On the other hand, it was indicated that copper’s higher level may enhance the genetic predisposition to increased CRP level, suggesting that copper exposure plays a role in triggering inflammation. In addition, higher zinc plasma concentration weakened the interaction between copper and CRP. Another curtail aspect is that copper and zinc are being absorbed from the small intestine and high doses of zinc administration can result in copper deficiency so appropriate dosage is fundamental. In the course of therapy, the possible level of zinc and copper in the serum should be checked first, as these parameters are not routinely determined in patients with COVID-19. Knowing concentrations of those microelements in serum, it is only possible to introduce the proper supplementation of zinc ana copper. Supplementation in patients with deficits in copper or zinc may positively influence COVID-19 disease course, as both increase the survival rate, immune response, antioxidative defense systems and antiviral properties.

## Figures and Tables

**Figure 1 biology-11-00752-f001:**
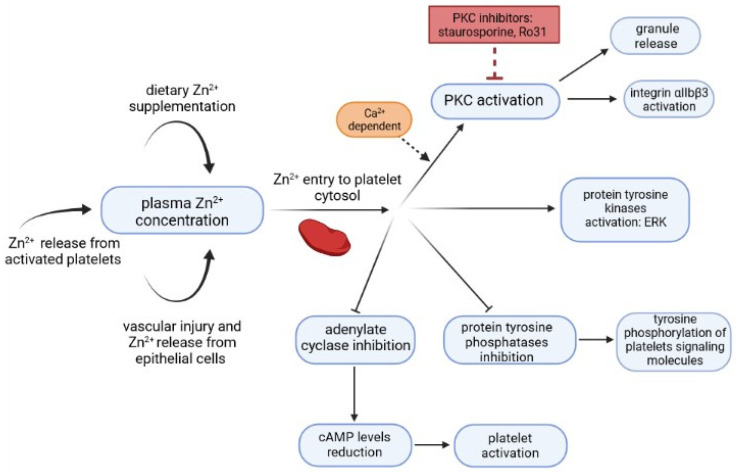
Diagram presenting the process of platelets activation caused by Zn^2+^ [7,44,45,46].

**Figure 2 biology-11-00752-f002:**
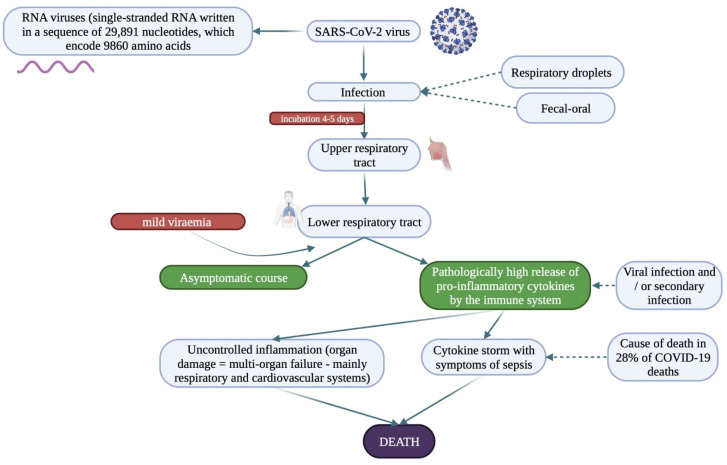
Diagram presenting the course of infection caused by the SARS-CoV-2 virus and possible options for the pathway of the disease process [101,103,107,108].

**Figure 3 biology-11-00752-f003:**
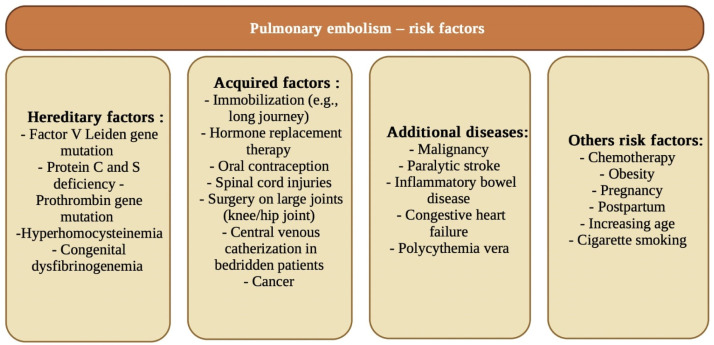
Risk factors predisposing to pulmonary embolism [114,115,116].

## Data Availability

Not applicable.

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
