# Peer review of "The Role of Zinc and Copper in Platelet Activation and Pathophysiological Thrombus Formation in Patients with Pulmonary Embolism in the Course of SARS-CoV-2 Infection"

_biology, 2022, doi:10.3390/biology11050752_

Round 1

Reviewer 1 Report

The authors presented a nice review concerning SARS-CoV-2 infection. In the last two years numerous reviews are published on this field and the authors focused on the role of zinc and copper on development of complications during SARS-CoV-2 infection. The title of this review is  “The role of zinc and copper in platelet activation and pathophysiological thrombus formation in patients with pulmonary embolism in the course of SARS-CoV-2 infection”. However, only a small part of the review is devoted to zinc and copper and most part of it connected with already well known (and described in many other reviews and original papers) mechanisms of SARS-CoV-2 infection. Numerous literature described the role of zinc and copper on platelet function and coagulations system, however the authors only very briefly analyzed this literature. The conclusion of the review that “Based on available data, we hypothesize that enrichment of plasma copper levels will increase immunity in people.” looks not logical because before they wrote that Cu deficiency is rare and usually only happens in seriously ill people receiving parenteral nutrition that lacks Cu”. Clearly, zinc and copper are not the major problem of SARS-CoV-2 infection and it is hard to expect that it will be organized “more studies and multi-centre prospective clinical trials in this area”

Some sentences should revised.

  1. 5, L. 160. Heat shock reactions (HSP) – chaperone proteins and proteases. HSP means “heat shock reactions proteins, not reactions and proteases is to broad, it should be more concrete.
  2. 5. L. 184. “it binds to it” should be revised.
  3. 5. L. 195. “globular subunits that contain amino acid residues” should be revised.
  4. 6. L. 204. “Moreover, by binding to its ligands CRP dissociates to monomers which present different functions compared to pCRP” Nothing was mentioned about ligands before.
  5. 6. L. 242. “Other proinflammatory properties of mCRP include platelet activation or thrombus formation. Platelet activation or thrombus formation are not proinflammatory properties.
  6. 7. L. 265. “As CRP expression in active plaques is increased” not expression, but amount.
  7. 7. L. 271-273. “However, the study showed that the platelet receptor glycoprotein IIb-IIIa (GPIIb-IIIa) blockage prevents the pCRP dissociation to mCRP. Thus, a reduction in platelet deposition at the arterial wall was obtained [83].” Not correct statement and citation (83). In the cited paper noting is mentioned about GPIIb-IIIa inhibition and CRP. It should be revised.
  8. 7. L. 280. “GPIb-IX-V complex, which is a potent platelet receptor.” This complex is a well known vWF receptor.
  9. 7. L. 286. “Exogenous Zn2+ may influence platelet activation and initiate its aggregatory mechanisms [98].” 98 is a nice review concerning effects of zinc on platelets and the authors focused on the main topic of their review more than just by one sentence.
  10. 8. L 346. “Zinc supplementation is considered to decrease mortality in patients with COVID-19 and to enhance cytotoxicity when used in vitro [117].” Ref. 117 is from 2014 and at that time there was no COVID-19!
  11. 8. L. 353-357 “There is a lot of evidence linking zinc to blood clotting which is important in con- junction with pulmonary embolism during SARS-CoV-2. Zinc is released from cells called platelets that control blood clotting, and scientists have found unwanted blood clots can form when zinc levels in the blood are faulty. Also, histidine-rich glycoprotein and fibrinogen, which are proteins in the blood that stop the action of natural anti-clotting factors called heparins, depend on zinc to work.” These sentences should be revised and citations are needed.
  12. The authors should rewrite the review with more focus on zinc and copper as it is stated on the title of this review.

Author Response

Aleksandra Górska
Medical University of Lublin
Department of Human Anatomy
Jaczewskiego 4
20-090 Lublin, Poland
[email protected]

Dear Reviewer,

Thank you very much for reviewing our manuscript. We appreciate the interest and commitment you have provided for this work. We are very grateful for your extremely precious comments. We are convinced that thanks to your suggestions this manuscript will be much more valuable.

We are pleased to submit explanations and details of our revisions in the manuscript entitled “The role of zinc and copper in platelet activation and pathophysiological thrombus formation in patients with pulmonary embolism in the course of SARS-CoV-2 infection”.

The followings are our point-by-point responses:

  1. L. 160 – the abbreviation HSP has been corrected i.e., Heat Shock Proteins. The division was modified to include two groups of Hsps.
  2. L. 184. “It binds to it” has been revised.
  3. L. 195 - “globular subunits that contain amino acid residues” has been revised. We clarified that each subunit consists of 206 amino acid residues.
  4. L. 204 – “Moreover, by binding to its ligands CRP dissociates to monomers which present different functions compared to pCRP” has been corrected to highlight the importance of CRP dissociation into the monomers.
  5. L. 242 – “Other proinflammatory properties of mCRP include platelet activation or thrombus formation” sentence was corrected and moved to more appropriate paragraph. Reference was changed to “37,80”.
  6. L. 265 – “As CRP expression in active plaques is increased” has been revised.
  7. L. 271-273 – “However, the study showed that the platelet receptor glycoprotein IIb-IIIa (GPIIb-IIIa) blockage prevents the pCRP dissociation to mCRP. Thus, a reduction in platelet deposition at the arterial wall was obtained [83]” has been revised and proper citation was added – the proper reference: 38.
  8. L. 280 – “GPIb-IX-V complex, which is a potent platelet receptor”. This information is supported by “Gresele P., Kleiman N.S., Lopez J.A., Page C.P. Platelets in Thrombotic and Non-Thrombotic Disorders. Pathophysiology, Pharmacology and Therapeutics: An Update”. We found this GPIb-IX-V complex property important for our work.
  9. L. 286 – “Exogenous Zn2+ may influence platelet activation and initiate its aggregatory mechanisms [98]” more information was added.
  10. L 346 – corrected.
  11. L. 353-357 – the sentence was revised.
  12. The paragraph on copper and zinc has been enriched with additional information.

We hope that after this revision, the manuscript is of a higher quality and worth reading.

We wish you all the best

Sincerely,

Aleksandra Górska
on behalf of all authors

Reviewer 2 Report

The review by Monika Szewc et al. concluded the pathophysiology of thrombus formation in patients with pulmonary embolism in the course of COVID-19 disease, the role of zinc and copper in the process, and further therapeutic solutions in COVID-19 treatment. It is an interesting and meaningful research review. The review presented are generally strong, and appear convincing, but would benefit with further to help strengthen the main conclusions and to better understand.

Major comments

  1. The author should add more graphic schematic diagrams in the paper for quickly understanding. Sometimes, too many text descriptions are not easy for readers.
  2. The figure 1 and Table 1 showed in this review have the poor quality.

Minor comments

  1. All across the review, author should uniformly use past tense or present tense when quoting the published data. About the same mistakes in the following, I won’t make notes.
  2. Line 86-89, “Infection of SARS-CoV-2 virus and hence the gradual destruction of lung cells in-86 duces a local immune response that results in the recruitment of macrophages and 87 monocytes which in response to infection release cytokines and stimulate the immune 88 response to produce T and B cells” This sentence is so long and hard to understand.
  3. Line164, “Varga et al. gave reported that” should be “Varga et al. reported that”, no gave.
  4. Line173, “In the other hand, zinc has a role in” should be “On the other hand,”.
  5. Line245-246, “which is a potent endotheli-245 um-derived constrictor and by altering renin-angiotensin system activation causing vas-246 oreactivity impairment” should be corrected, the sentence is not right.
  6. Line249, “between metals and CRP, which levels” should be “whose levels”.
  7. Line271, “growing platelet aggregate where it stimulates further platelet deposition” should be “aggregation”.
  8. Line281-283, “It was supported by the fact that one of its major ligand-binding subunits which is glycoprotein Ibα, was directly binded by mCRP [95].” It is too complicated, not good for reading.

Author Response

Aleksandra Górska
Medical University of Lublin
Department of Human Anatomy
Jaczewskiego 4
20-090 Lublin, Poland
aleksandragó[email protected]

Dear Reviewer,

Thank you very much for reviewing our manuscript. We appreciate the interest and commitment you have provided for this work. We are very grateful for your extremely precious comments. We are convinced that thanks to your suggestions this manuscript will be much more valuable.

We are pleased to submit explanations and details of our revisions in the manuscript entitled “The role of zinc and copper in platelet activation and pathophysiological thrombus formation in patients with pulmonary embolism in the course of SARS-CoV-2 infection”.

The followings are our point-by-point responses:

Major comments:

  1. We created new graphics. If in the opinion of the reviewers, more are needed, because some content will be incomprehensible, we are open to suggestions. The new graphics have been created using a special program and are in jpg format, so that the quality is at its best.
  2. Figure 1 has been corrected i.e., the case has been increased, the file format has been changed from png to jpg which significantly increased the quality. Table 1 was re-created in another program and its format was modified to jpg.

Minor comments:

  1. We have tried to standardize the time used when citing.
  2. Line 86-89 – the sentences have been broken down to make them more understandable and enhanced with minor information.
  3. Line164 – corrected.
  4. Line173 – corrected.
  5. Line245-246 – corrected.
  6. Line249 – corrected.
  7. Line271 – corrected.
  8. Line281-283 – corrected.

We hope that after this revision, the manuscript is of a higher quality and worth reading.

We wish you all the best

Sincerely,
Aleksandra Górska
on behalf of all authors

Round 2

Reviewer 1 Report

The manuscript is improved accordingly, I have no more comments.